# The Present Utility of the Oxytocin Challenge Test—A Single-Center Study

**DOI:** 10.3390/jcm9010131

**Published:** 2020-01-03

**Authors:** Anna Różańska-Walędziak, Krzysztof Czajkowski, Maciej Walędziak, Justyna Teliga-Czajkowska

**Affiliations:** 12nd Department of Obstetrics and Gynecology, Medical University of Warsaw, 00-315 Warsaw, Poland; aniaroza@tlen.pl (A.R.-W.); krzysztof.czajkowski@wum.edu.pl (K.C.); justyna.teliga@gmail.com (J.T.-C.); 2Department of General, Oncological, Metabolic and Thoracic Surgery, Military Institute of Medicine, 04-141 Warsaw, Poland

**Keywords:** oxytocin challenge test, fetal heart rate, cesarean section, pregnancy, cardiotocography

## Abstract

Introduction: The oxytocin challenge test (OCT) used to be one of the most important tools in assessing fetal well-being before ultrasonography became prevalent. We show that, after modifying the classification of the results and the intervention algorithm, OCT can still be a useful tool in present-day obstetrics. Material and methods: The study included 318 OCTs performed in patients admitted to our department from 2010 to 2012. A modified classification of test results was introduced, dividing the results in four groups: I-negative, II-positive, III-non-diagnostic and type IV (fetal tachycardia or increased variability). The purpose of the study was to evaluate the clinical significance of OCT in assessing intrauterinal fetal well-being and predicting the necessity for ending the pregnancy. Results: A significant difference (*p* < 0.001) in the delivery method and the indications for cesarean sections (CS) was found between negative and positive OCT results. CS indicated by an abnormal fetal heart rate (FHR) pattern had to be performed in 40% of cases with positive OCT results, having constituted 84.6% of all CSs in this group. After negative OCTs, 12.8% pregnancies were ended by CS from FHR indications (62.3% of all the indications). Conclusions: A positive OCT result can be a valuable predictor of an abnormal fetal heart rate pattern after the test and during the delivery, as well as a higher probability of a CS from cardiotocography (CTG) indications, with positive predictive value (PPV) 0.50 and negative predictive value (NPV) 0.85.

## 1. Introduction

The oxytocin challenge test (OCT) is a diagnostic tool that can be used in case of suspected placental insufficiency to evaluate the fetal tolerance of uterine contractions. The history of OCT goes back to 1960s, when Hammacher invented the idea of OCT, further introduced and developed by Caldeyro-Garcia and Pose [1]. Oxytocin given intravenously induces uterine contractions that cause intermittent decrease in oxygen flow to the fetus, which allows clinicians to determine cases with previously deteriorated oxygen flow. The decrease in blood oxygen levels during contractions results in decelerations in fetal heart rate (FHR) [2].

The historical classification of OCT results included five categories. A negative test was defined by the presence of accelerations in the fetal heart rate, lack of decelerations and normal variability. A positive result was defined by late decelerations after more than 50% of contractions. A suspicious result included all the remaining tests with decelerations present. Equivocal or suspicious results included the remaining tests with decelerations other than late and repetitive. The classification also recognized hyperstimulation tests with contractions more frequent than one per two minutes, lasting more than 90 s and non-diagnostic tests with the absence of contractions or low FHR variability. In the beginnings of OCT, ultrasonography was still developing and cardiotocography (CTG) with OCT were the most important tools in assessing fetal intrauterine well-being. The diagnostic value of ultrasonography increased over the years and it has become the most important diagnostic tool in present-day obstetrics together with CTG monitoring. According to the Society of Obstetricians and Gynecologists of Canada Guideline, there is still a place for OCT in a modern obstetric unit. It should be considered in case of pathological or suspicious non-stress test (NST) to evaluate uteroplacental function during contractions in patients primarily qualified for a vaginal delivery [3]. As OCT has been abandoned in many perinatal centers, in our study we wanted to present the still-existing utility of OCT in modern obstetrics, with an additional upgrade in the form of a modified classification of the test results.

## 2. Material and Methods

Our retrospective study includes all the OCTs performed in a single tertiary perinatal center between 2010 and 2012, which constituted 318 tests. The study group comprised 318 patients with singleton pregnancies hospitalized in our clinic. The exclusion criteria comprised multiple pregnancies, presence of placenta previa, dehiscence of the uterine scar after previous cesarean sections (CS), less than 34 weeks of pregnancy and lack of consent for the test. The inclusion criteria comprised all the indications mentioned below.

The indications for OCT were divided into the following categories: intrauterine fetal growth restriction (IUGR), pre-pregnancy medical history (diabetes mellitus, pre-pregnancy hypertension, thrombophilia, etc.), pregnancy-induced medical conditions (gestational diabetes mellitus, pregnancy-induced hypertension, intrahepatic cholestasis of pregnancy, etc.), adverse obstetric history, pregnancy after term, any previous suspicious CTG tracings, abnormal fetal movements and increased or reduced amniotic fluid volume. All the remaining indications were included into ‘other’ subgroup.

The purpose of the study was to find a correlation between the type of OCT results and the following pathological CTG tracings, means of delivery, indications to CS and the newborn’s condition in order to establish the utility of modified OCT in deciding about ending the pregnancy.

OCT was performed with oxytocin given in continuous intravenous flow (5 IU in 500 mL of compound electrolyte solution), starting with 0.05 mIU/min. The flow velocity was increased by 0.05 mIU/min every 15 min to achieve 3 uterine contractions per 10 min and then it was continued for at least 45 min. The flow was discontinued in case of a deceleration of FHR longer than 1 min of duration or repetitive shorter decelerations. A deceleration was defined as a decrease in FHR of equal or more than 15 beats per minute (bpm), lasting for at least 10 s; early decelerations were defined as starting before the peak of contraction; late decelerations, starting after the peak of contraction; variable decelerations, with no correlation with the uterine contractions. Fetal tachycardia was defined as an acceleration in FHR of more than 160 bpm lasting for 10 min or more. Reduced variability was recognized in cases of variability of fewer than 10 bpm for more than 20 min; increased variability if more than 25 bpm. If the patient did not have continuous CTG monitoring administered for different reasons, the CTG recording was ended 30 min after the end of contractions.

We analyzed the general and obstetric history as well as CTG tracings of 318 patients included in the study, implementing the data obtained in a database created with Microsoft Access. The analyzed CTG data included the computerized CTG recordings from before, during and after OCT tests, as well as during the delivery, and were archived on Phillips OB Tracevue servers in our clinic. For the purpose of this study, CTG tracings were divided into four categories: (1) normal, (2) intermittent abnormalities, (3) frequent abnormalities, and (4) qualifying to end the pregnancy. The newborns’ condition was evaluated with the Apgar score.

We modified the existing classification of OCT results, dividing them into four types as follows:(I)Negative: no decelerations, normal baseline and variability(II)Positive: any decelerations present in the FHR pattern(III)Non-diagnostic: with periods of reduced variability longer than 20 min(IV)Episodes of fetal tachycardia or high variability longer than 20 min

The modified classification of OCT results is used in our department and throughout our study to assess all the tests performed.

All of the participants provided consent prior to inclusion and approval by the ethics committee has been obtained from Institutional Review Board (AKBE/118/14).

Bias: as the study was conducted in a tertiary perinatal center, the percentage of patients with high-risk pregnancies was higher than in general population which can be considered as a limitation to the study, the only subgroups of potentially healthy individuals being the group of abnormal fetal movements and pregnancy after term. As interpretation of CTG tracings is a subjective method of evaluation, we tried to decrease the level of possible bias by following the established criteria of OCT results and having the test results assessed by two independent specialists in obstetrics. In case of differences, the results were consulted with a specialist in perinatology. Nevertheless, evaluation bias cannot be completely excluded.

We performed the statistical analysis using StatSoft Statistica version 6.1 PL (StatSoft Inc., Tulsa, OK, USA). Normality of the data was tested with Shapiro–Wilk test. Continuous variables were compared with the Student’s *t* test for normally distributed or Mann–Whitney U test for non-normally distributed data. Categorical variables were compared using the chi^2^ or Fisher test. Statistical significance was set at *p* < 0.05.

## 3. Results

The average weight gain during pregnancy in our study group was 13.45 kg, the average number of pregnancies (with inclusion of the present pregnancy) was 1.91 ± 1.1, the average gestational age at the moment of OCT 260.90 ± 11.7 days. Table 1 presents the baseline characteristics of the population.

The characteristics of the test results depending on the indications is shown in Table 2 (Chi sq. = 26,461, df = 8, *p* < 0.001). An increased number of positive results was observed in patients qualified to OCT because of IUGR (38.3%) and suspicious CTG recordings before the test (48.4%). In the groups of pre-pregnancy and pregnancy-induced medical conditions the rate of positive results was significantly lower, respectively 22.2% and 15.8%. The number of patients in groups III and IV was too low to perform statistical analysis.

There was a significant correlation found between the test results and the method of delivery (Table 3) (Chi-sq. = 63.427, df = 16, *p* < 0.001). In the group of 204 patients with negative OCT results, 131 (64.2%) had a spontaneous vaginal delivery, 26 (12.8%) a CS due to pathological CTG trackings, 43 (21.1%) a CS due to other indications and 4 (2.0%) a vacuum delivery. In the group of 90 patients with positive OCT results, 37 (41.1%) had a spontaneous vaginal delivery, 36 (40.0%) a CS due to pathological CTG tracings, 8 (8.9%) a CS due to other pre-existing indications and the positive result of OCT, 8 (8.9%) a cesarean section due to other indications and 1 (1.1%) a vacuum delivery. The number of patients in groups III and IV was too low to perform statistical analysis, therefore the results from those two groups were not considered in the further analysis.

The rate of operative interventions due to suspicious or pathological CTG tracings following the test in the group II was significantly higher than in the group I (50.0% vs. 14.7%). The total rate of CS in the group II was higher (57.8%) compared to the average rate of CS in the analyzed time period in the Mazovian province (comparable in our department), given at about 34.0% [4].

If the positive predictive value (PPV) of OCT is considered as the number of pregnancies that had to be ended because of suspicious or pathological CTG tracings, PPV of the positive result of the test is 50%. If the negative predictive value (NPV) is considered as the number of pregnancies that ended without the necessity for operative intervention due to CTG trackings, the NPV of the negative result of OCT is 85%.

To evaluate the condition of the newborns, the Apgar score was used, with no significant differences between the groups of test results (Chi-sq. = 1.94, df = 3, *p* < 0.585).

## 4. Discussion

There are several methods of antepartum testing, including non-stress test (NST), ultrasonography, and OCT [5]. As stated by Freeman et al. in a multi-institutional study which included 1542 NSTs and 4626 OCTs, a negative result of OCT is considered to lead to a low rate of 0.04% of stillbirth within a week after the test, compared to 0.32% in pregnancies monitored with NST only [6]. In our study we did not observe any stillbirths.

Numerous institutions, including our center, consider OCT to be a useful tool in cases of suspected IUGR to establish the optimum time and mode of delivery and therefore reduce the number of unnecessary intrapartum CSs, the number of which is becoming epidemic. Yefet et al. analyzed 354 OCTs performed because of suspected IUGR [7]. The purpose of the study was to evaluate the utility of OCT as a method of reducing the number of intrapartum CSs performed due to pathological CTG tracings. The pregnancy was ended with CS in case of positive or ambiguous results or labor was induced in case of negative results. The IUGR was affirmed in 264 cases (all 90 cases without confirmed IUGR had negative test results). The number of intrapartum CSs due to pathological CTG tracings was comparable in the cases of IUGR with negative results of OCT and non-IUGR cases.

Tanaka et al. presented a case–control retrospective study about the efficacy of OCT in management of IUGR, having included 73 cases [8]. They found significant differences in oxygen saturation in the umbilical artery (UA) between the groups and concluded that OCT allows early intervention in cases of IUGR, which enables fetal acidemia and acidosis to be avoided. Fetuses with IUGR would respond to contractions with decelerations due to reduced resources provided by an impaired blood flow, which was also confirmed in our study with a significantly higher percentage of positive results when IUGR was an indication to OCT. In our study, we observed only a few cases of fetal acidemia and acidosis and the vast majority of neonates were born in good or very good condition according to blood pH, CO_2_ blood concentration and Apgar score.

Fu and Olofsson analyzed a group of 126 patients with suspected IUGR in order to find a correlation between fetal brain-sparing circulation brain-sparing flow (BSF), OCT, mode of delivery and fetal outcome [9]. BSF was defined as a middle cerebral artery-to-umbilical artery pulsatility index ratio of <1.08. OCT was performed to determine the timing and mode of delivery, after exclusion of cases with pathological CTG tracings or absent or reversed end-diastolic flow in UA, that were routinely qualified to CS. The OCT-negative cases had their pregnancy continued under high surveillance or labor induced, the OCT-positive cases were delivered by CS. The presence of BSF was established as a poor predictor of a positive OCT result and of CS in OCT-negative cases. Fu and Olofsson presented the utility of OCT as tool allowing to decision-making about CS in cases of IUGR. In our study, patients were qualified for OCT not only because of suspected IUGR, but also for different indications and we did not consider a positive result of OCT as a direct indication to perform immediate CS. The mode of delivery was chosen depending on the indications for OCT, the CTG recordings before and after the test and Doppler ultrasound examinations, therefore there were more cases of labor induced.

Li, Gudmundsson and Olofsson carried out several studies analyzing the fetal and maternal blood flow during OCT contractions. Olofsson proved there was a correlation between decelerations in the fetal heart rate and a rise in vascular resistance in UA and uterine arteries [10]. In another study Li, Gudmudsson and Olofsson described an augmented pulsation index (PI) in UA during OCT only in the group of patients with positive test results compared to no increase found in the OCT-negative cases [11]. No correlation was found between the increased PI in UA and fetal negative outcome or the ratio of intrapartum CSs performed due to pathological CTG tracings.

Li, Gudmundsson and Olofsson showed an increased ratio of BSF in the group with IUGR and positive OCT results [12]. They analyzed 82 cases, having performed ultrasound Doppler examination of the fetal blood flow in middle cerebral artery (MCA) and UA before and during the test, in the course of contraction and relaxation of the uterus, along with continuous CTG analysis. The PI ratio in UA during contraction and relaxation proved to be significantly higher in the OCT-positive result group. The dip in MCA resistance index was significantly more marked in the OCT-positive group as well.

Figueras et al. compared the predictive value of OCT and Doppler ultrasound evaluation of ductus venosus (DV) flow in fetuses with IUGR, BSF and non-reactive NST [13]. There were 68 patients included in the study, the end-points being a pH < 7.10 in umbilical cord blood, neonatal intensive care unit (NICU) admission, intubation and severe neonatal morbidity. PPV of DV blood flow Doppler ultrasound examination was found to be higher than of OCT and abnormal blood flow in DV was correlated with poor neonatal outcomes. As stated before, the vast majority of neonates in our study were born in good or very good condition, therefore our results cannot be compared.

## 5. Conclusions

The oxytocin challenge test can still be used in modern obstetric units as a reliable tool in high-risk pregnancies to evaluate potential placental insufficiency and fetal well-being, complementary with CTG analysis and ultrasound examination. We recommend using our modified classification of OCT results as it is more justified in current perinatology. A positive result of OCT predicts the possible necessity of an earlier delivery and a negative result enables the obstetrician to decide about prolongation of the pregnancy while carefully monitoring fetal well-being.

## Figures and Tables

**Table 1 jcm-09-00131-t001:** Baseline characteristics.

	Mean	df
Age (years)	30.80	4.34
Height (m)	1.67	6.16
Weight (kg)	66.60	14.32
Weight gain (kg)	13.45	5.60
Total pregnancy length (days)	265.78	11.42

**Table 2 jcm-09-00131-t002:** Characteristics of the test results depending on indications to oxytocin challenge test.

Indications	Test Result (*n* = 318)			
	I Negative (*n* = 204)	II Positive (*n* = 90)	III Non-Diagnostic (*n* = 8)	IV Fetal Tachycardia or High Variability > 20 min (*n* = 16)
Pre-pregnancy medical history (*n* = 90)	58 (64.4%)	20 (22.2%)	5 (5.6%)	7 (7.8%)
Pregnancy-induced medical conditions (*n* = 57)	46 (80.7%)	9 (15.8%)	1 (1.8%)	1 (1.8%)
Intrauterine fetal growth restriction (*n* = 47)	28 (59.8%)	18 (38.3%)	0	1 (2.1%)
Abnormal amniotic fluid volume (*n* = 8)	5 (62.5%)	2 (25.0%)	0	1 (12.5%)
Adverse obstetric history (*n* = 19)	15 (79.0%)	4 (21.1%)	0	0
Pregnancy after term (*n* = 16)	12 (75.0%)	4 (25.0%)	0	0
Suspicious cardiotocography (*n* = 64)	26 (40.6%)	31 (48.4%)	2 (3.1%)	5 (7.8%)
Abnormal fetal movements (*n* = 10)	8 (80.0%)	1 (10.0%)	0	1 (10.0%)
Other (*n* = 7)	6 (85.7%)	1 (14.3%)	0	0

**Table 3 jcm-09-00131-t003:** The oxytocin challenge test (OCT) results and the method of delivery.

Test Result/Method of Delivery	Vaginal Spontaneous Delivery	Cesarean Section (Pathological Cardiotocography Tracings)	Cesarean Section (Suspicious Cardiotocography Tracings During OCT)	Cesarean Section (Other Indications)	Vacuum Extraction
I Negative (*n* = 204)	131 (64.2%)	26 (12.8%)	0	43 (21.1%)	4 (2.0%)
II Positive (*n* = 90)	37 (41.1%)	36 (40.0%)	8 (8.9%)	8 (8.9%)	1 (1.1%)
III Non-diagnostic (*n* = 8)	4 (50.0%)	1 (12.5%)	1 (12.5%)	2 (25.0%)	0
IV Fetal tachycardia or high variability > 20 min (*n* = 16)	7 (43.8%)	5 (31.2%)	0	4 (25.0%)	0

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
