# Peer review of "The Present Utility of the Oxytocin Challenge Test—A Single-Center Study"

_jcm, 2020, doi:10.3390/jcm9010131_

Round 1

Reviewer 1 Report

jcm-660003

Anna Różańska-Walędziak , Krzysztof Czajkowski , Maciej Walędziak * , Justyna Teliga-Czajkowska

The oxytocin challenge test – is it still justified in modern obstetrics?

Thank you for the opportunity to review this manuscript in which the authors aim to demonstrate that the OCT tests has some utility in modern obstetrics, although I am not clear what would its utilization. Initial reading of the title created the impression that this might be a review. I suggest to the authors to change the title from the a question to a statement representative for their study. 

The grammar and sentences structure of this manuscript needs corrections. In addition, the medical terminology appears to be translated in English instead of using the corresponding English medical terminology. Overall, is a manuscript difficult to read and understand. The material is not well organized, with interchanging information between manuscript sections. The outcomes of the study are not compelling to support the utility of OCT in modern obstetrics. Below I have included only few specific comments to exemplify the areas that need more work.

The rationale for this study is not convincing or supported enough by the evidence provided by the authors. The authors themselves acknowledge that US provide all the information and more than the OCT tests can provide. Furthermore, the US measurements are standardized and reliable in comparison with the CTG tracing whose interpretation varies more with the individual who reads the tracing. Dawson-Redman CTG criteria would help with the decision for delivery. The OCT test is invasive and likely does not have a place in modern obstetrics. The authors indirectly confirm this, as their data is old (2010). 

The aim of the study is a general statement, “The purpose of the study was to evaluate whether OCT can be a valuable tool of diagnosis in modern obstetric units”.  Diagnosis tool of what?

Methods section does not follow a plan of presentation and appear overall messy. By the end of the section the reader is still not clear of the study aim, the population or the outcomes. The description of statistical methods is also lacking clarity; it appears that the authors have named some tests but is not clear what and how was used. Not clear how the OCT tests outcomes were classified in analysis. However, the methodology is inadequate, and the results obtain using this methodology does not answer the study question.

Results section contain paragraphs that explain the methodology. In addition, the definition of the nine categories of OCT indications is unclear and not well done. For instance, what is a  “grave medical history”? What the authors means by “history or increased risk of thrombosis”? Almost all categories defined in this paragraph are obscure and incorrect. Table 1 does not reflect this classification but shows the characteristic of the population.

The authors state: “An increased number of positive results was observed in patients qualified to OCT because of fetal hypotrophy (38.3%) and suspicious CTG recordings before the test (48.4%)”. These findings could have been predicted from before doing the OCT: the IUGR/FGR babies (fetal hypotrophy is a term no longer in use; plus, it was not defined) have poor resources and will respond with decelerations to uterine contractions. Furthermore, if there is an abnormal CTG in absence of labour/contractions shows that there is fetal and/or placental pathology. The authors themselves acknowledge in the Discussion section that the value of OCT is inferior in predicting fetal outcomes in comparison with US/Doppler.

Discussion section has several paragraphs with subheadings that are not clear, i.e Research Implications. Furthermore, many sentences are grammatically incorrect and difficult to read and understand. Interestingly enough, the authors discuss stillbirth in the second paragraph of this section, a term that has not been mentioned until here. Most of this section is dedicated to describing in detail other studies, without making a parallel or showing the significance for the present study.

Overall, the tables are modest. 

Author Response

Dear Reviewer 1,

We respectfully read your analysis of our manuscript and revised our manuscript accordingly. Thank you for your valuable remarks.

According to your suggestions, we had the manuscript reviewed by an English speaker.

We changed the title from a question to a statement representative for our study.

We reorganized the sections of the manuscript in order to avoid repetitive information.

We tried to present the outcome of the study in a way to support our conclusion about still-existing utility of OCT.

We acknowledge the dominant role of ultrasonography and cardiotocography. To reduce the risk connected with conducting the test, we created the exclusion criteria and changed the definition of a positive test result, considering any decelerations as a positive result of the test. Obviously, OCT still remains dependable on the individual medical professional assessing the CTG tracings and cannot be compared with ultrasonography standards.

We proposed OCT as an additional diagnostic tool that allows to choose the optimum time of ending the pregnancy in ambiguous cases, as well as helping with the decision about i labor or performing a cesarean section.

We reorganized the methods section and cleared the classification of the test results. We moved the section about the OCT indication to the methodology paragraph and tried to specify the classification of indications. Thrombophilias were considered as higher risk of thrombosis.

The discussion paragraph was reorganized according to your suggestion.

Reviewer 2 Report

The scientific work is well conducted, but it has to be improved in the presentation of the data and results, because they are not so clear to the readers. The number of patients is low for the aim of the work. Otherwise it is still interesting to discuss of OCT today, and it makes the originality of the scientific work. It needs some changes in the English style, but overall it is a good scientific work.

Author Response

Dear Reviewer 2,

Thank you for your positive opinion and assessment of our manuscript.

According to your remarks, we had the manuscript reviewed by an English speaker.

We tried to ameliorate the presentation of the data and the results by clearing the classification of the test results used in our study, as well as the subgroups of indications for OCT.  We hope it will improve our manuscript.

Unfortunately, we are not able now to include more patients in the study group.

Reviewer 3 Report

I agree with the authors that there is a place for the OCT (CST) in modern obstetrics. I am not sure tat this article in its current state helps.

If the authors are going to introduce a new classification system for grading OCT's , then they should be clear with Type I, II, III and IV, rather than using positive, negative, nondiagnostic and then type IV.

I understand that some of the exclusion categories were related to contractions being relatively contraindicated, but unclear why <34 weeks is an exclusion.

I am not familiar with the term fetal hypotrophy. Is this equivalent to growth restriction or is it some other term.

Pitocin dosing is usually in miu/min, rather than ml/hour. Not familiar with the term blood gasometry.

Author Response

Thank you for your thorough analysis of our work, we revised the manuscript accordingly. According to your remarks, we had the manuscript reviewed by an English speaker.

We hope that together with additional corrections it will ameliorate the quality of the article.

We corrected the classification of OCT results according to your suggestion and cleared into types I, II, III and IV throughout the whole manuscript, including the tables.

In our department <34 weeks of pregnancy is an exclusion to OCT as it is considered a risk factor of preterm labor.

We exchanged the term ‘fetal hypotrophy’ for the intrauterinal growth restriction.

We corrected the oxytocin dosing into mi.u./mmin. We removed the term blood gasometry.

Round 2

Reviewer 1 Report

Thank you for the opportunity to review once again the manuscript with the original title

“The oxytocin challenge test – is it still justified in modern obstetrics?”.

Overall, the changes made by the authors to the manuscript were minor. The most important issues have not been addressed. Moving sentences around does not qualify as a revision. English grammar and sentence structure still require improvement. The new title suggests a review and not an original study. 

Specific comments are included below:

Introduction

There is no statement/argumentation for what the need for this study. The last paragraph of an Introduction of any scientific paper should identify the aim of the study and the novel approach to the subject, if any. The sentences from the last paragraph are confusing and is not clear what the message are they conveying.

Methods

This section is still missing essential information.

The study population is not defined, and the recruitment is not clear. From where this population was recruited, how, who, why? What are the inclusion criteria? What CTG machines did you use? The characteristics of the recordings (speed, paper, computerised or not etc ) including the length of time, who did the recording, where, who read the recordings etc. The definitions used for CTG interpretations are only partial. How the Oxytocin dose for OCT was chosen? The statement “The newborns’ condition was defined with the Apgar score” is unfortunate.

Statistics:

Based on the description provided, the authors themselves are not clear with what tests were used and why. Shapiro Wilk test is a test of normality. Hence, not clear how the authors used this test for assessing the continuous variable. This should be clarified. The statement “Other tests used in the analysis were the Chi137 square, Mann-Whitney and the Kruskal-Wallis test” does not suffice. For what these tests were used and how must be included.

Results

The results do not have a Table 1 which always include a description of the population.  This section still includes information that should be in the Methods section.

Discussion

Few sentences added by the authors do not improve its quality. Some of those sentences contain grammar errors and do not really add anything to the section.

Author Response

Dear Reviewer 1,

Thank you for your review of our manuscript and all the valuable remarks.

Title

We changed the title into a one suggesting an original study.

Introduction

As you suggested, we reduced and cleared the last paragraph of the introduction section and added a sentence about the aim and novelity of our study at the and of introduction section.

Methods

In the methods sections, we tried to clearly present the recruited population. The inclusion criteria comprised all the indications mentioned in the further fragment of the manuscript.

As it is described in the paragraph, we use Philips OB Tracevue system (computerized), we did not use paper recordings and all the data are archived on our computer servers.

OCT was performed with oxytocin given in continuous intravenous flow, starting with 0.05mi.u/min with no differences for all the patients. The flow velocity was increased by 0.05mi.u/min every 15 minutes to achieve 3 uterine contractions per 10minutes and then it was continued for at least 45 minutes. The only difference to be found between the patients is the time from the beginning of the oxytocin flow to the moment of achieving 3 contractions per ten minutes and this period is not considered yet as OCT. If the patient did not have continuous CTG monitoring, the CTG recording was ended 30 minutes after the end of contractions. The CTG recording was connected by a midwife and continuously analyzed by a specialist in obstetrics.

In our center, we use continuous CT monitoring, monitoring 6 times/24 hours for 60 minutes or for 20 minutes and 4 times/24 hours for twenty minutes depending on the patient’s condition and diagnosis. The form of monitoring used does not influence qualifying for OCT.

As we mentioned in our manuscript, the test results were assessed by two independent specialists in obstetrics. In case of differences, the results were consulted with a specialist in perinatology.

We corrected the sentence about the newborns’ condition.

Statistics

We corrected and clarified the information about types tests that were used and their purpose.

Results

We added a Table 1 with the description of the population in the study group.

We moved the paragraph about the bias of our study from the results to the material and methods paragraph.

Discussion

We tried to improve the quality of the discussion and correct the grammar errors.

Reviewer 2 Report

The modifications of the scientific work have improved it, so now it is ready for publication 

Author Response

thank you very much